# Recent Trends in Accounting and Information System Research: A Literature Review Using Textual Analysis Tools

**Fábio Albuquerque** [1,]* and **Paula Gomes Dos Santos** [1,2]

1  ISCAL—Lisbon Accounting and Business School, Instituto Politécnico de Lisboa,
   1069-035 Lisbon, Portugal; pasantos@iscal.ipl.pt
2  COMEGI, Universidades Lusíada, 1300-001 Lisbon, Portugal
*  Correspondence: fhalbuquerque@iscal.ipl.pt

**Abstract:** Accounting has been evolving to follow the latest economic, political, social, and technological developments. Therefore, there is a need for researchers to also include in their research agenda the emerging topics in the accounting area. This exploratory paper selects technological matters in accounting as its research object, proposing a literature review that uses archival research as a method and content analysis as a technique. Using different tools for the assessment of qualitative data, this content analysis provides a summary of those papers, such as their main topics, most frequent words, and cluster analysis. A top journal was used as the source of information, namely *The International Journal of Accounting Information Systems*, given its scope, which links accounting and technological matters. Data from 2000 to 2022 was selected to provide an evolutive analysis since the beginning of this century, with a particular focus on the latest period. The findings indicate that the recent discussions and trending topics in accounting, including matters such as international regulation, the sustainable perspective in accounting, as well as new methods, channels, and processes for improving the entities' auditing and reporting, have increased their relevance and influence, enriching the debate and future perspectives in combination with the use of new technologies. Therefore, this seems to be a path to follow as an avenue for future research. Notwithstanding, emerging technologies as a research topic seem to be slower or less evident than their apparent development in the accounting area. The findings from this paper are limited to a single journal and, therefore, this limitation must be considered in the context of those conclusions. Notwithstanding, its proposed analysis may contribute to the profession, academia, and the scientific community overall, enabling the identification of the state of the art of literature in the technological area of accounting.

**Keywords:** accounting; literature review; technology; textual analysis; *IJAIS*

## 1. Introduction

Accounting, as an applied social science, requires a constant analysis of the threats and opportunities that emerge, among others, in the political, social, environmental, economic, and legal contexts, which, in turn, are interconnected. The difficulty of adequacy and monitoring the developments associated with educational institutions' advent of new technologies has been stressed by literature [1]. It has also been demonstrated that research in accounting needs to evolve and follow contemporary trends that are present not only in the technological environment but also in other areas, including those related to other scientific fields [2]. Accounting research, according to Kaplan [3], has neglected topics of interest, presenting them as too rigid, cautious, and conservative. Additionally, scholars have identified a greater difficulty in publishing in this area compared to other areas of business, in general [4].

Although the definition of accounting has evolved, this is not reflected in the understanding of accounting as a social and moral practice, but only as a technical one, despite the efforts of academics in this regard over the last four decades [5]. Research is thus an

essential element for understanding the new forms of management in the so-called digital economy, as well as identifying the new skills and instruments that professionals must have to remain relevant and able to add value to the work they develop [6].

This paper performs a literature review (bibliometric analysis) of a top journal in accounting information systems (AIS) as a target, namely *The International Journal of Accounting Information Systems* (*IJAIS*). Papers released from 2000 to 2022 were gathered, aiming to present an evolutionary analysis of the recent trends in different areas of accounting research, with a particular focus on the latest period (from 2020 to 2022). Therefore, this paper's content analysis provides an overview of the main topics from the investigations covered by this journal over this period. Using different tools based on textual analysis, this exploratory paper offers a summary of the themes covered by those papers through different tools available for the assessment of qualitative data, such as their main topics, most frequent words, and cluster analysis. Their impacts and possible relationships are also included in this context.

Based on the methodology proposed, this paper may contribute to the literature by providing an overall perspective of the latest two decades of accounting literature in the technological field through a methodological approach that can be used in future investigations. To this point, topics considered as future avenues are discussed as opportunities in different fields for researchers. Research gaps in the literature are pointed out, which may be relevant for educational and professional practice in accounting. Consequently, this paper can contribute to professionals, academia, and the scientific community overall, enabling the identification of the state of the art in the accounting research area.

More specifically, the evidence found in this paper suggests the increasing use of emerging technologies in accounting and auditing in the latest years, with new research being developed through experimental and case studies. This can be seen as an open opportunity for researchers, academia, and practitioners since the gap between their different technological development levels, as stressed by the literature [1], may be reduced through closer projects and collaborations. The idea of a more conservative approach in accounting research [2–5] also appears to be mitigated in recent years through the findings from this paper that suggest the inclusion of a more diverse set of topics in the scope of the developed research in accounting. This is evidenced either by integrating trending topics, such as sustainability matters–also known as nonfinancial information or environmental, social and governance (ESG) issues–or by linking accounting as applied social science with other different social scientific areas, such as sociology and psychology from its perspective.

Literature reviews on the so-called emerging technologies in accounting have increased during the latest years, covering different periods, journals, and themes under assessment. For instance, research can be found on specific topics such as the use and implications of blockchains in accounting and auditing [7–10]. Moreover, specific accounting areas have also been selected for this purpose, such as the research by Mugwira [11] on auditing regarding internet-related technologies or neurosciences in accounting by Tank and Farrell [12].

Wider research on the use of technologies in accounting was performed by Kumar [13], Kroon et al. [14] and Chiu et al. [15]. Despite the focus by Kumar [13] on the research released in *IJAIS*, as well as the inclusion of this journal in the scope of the analysis performed by Kroon et al. [14] and Chiu et al. [15], those bibliometric analyses do not cover the latest period proposed in this paper. Therefore, considering this gap and the newest challenges accounting has been facing, it seems relevant to assess what issues and topics have been released by a top journal focused on the use of technologies in this area.

The structure of this paper is as follows: The next section provides the objectives and the methodology used, also reviewing the literature as previously proposed, and the third discusses the findings, also suggesting future avenues for similar research in this field and presenting its main limitations.

## 2. Literature Analysis

This section summarizes the papers published in the selected journal. It is divided into two subsections. The first provides an overview of the basic materials and methods proposed for the papers' analyses. The next one reports the results from the literature review.

### 2.1. Materials and Methods

This paper uses a qualitative and exploratory approach based on the archival research method and content analysis as a technique. The content within the abstract of papers released in the *IJAIS* was selected as the object of analysis since it represents a top accounting journal dedicated to information systems matters in this area. Those papers were gathered from the Scopus database by 30 September 2022.

Based on the information available on the Scopus website, "the *IJAIS* publishes thoughtful, well-developed papers that examine the rapidly evolving relationship between accounting and information technology (IT). Papers may range from empirical to analytical, from practice-based to the development of new techniques but must be related to problems facing the integration of accounting and IT. The journal addresses (but will not limit itself to) the following specific issues: control and auditability of information systems; management of IT; artificial intelligence research in accounting; development issues in accounting and information systems; human factors issues related to IT; development of theories related to IT; methodological issues in IT research; information systems validation; human-computer interaction research in AIS".

The *IJAIS* is classified as a Q1 journal in 2021, with a SCImago Journal Rank (SJR) of 1.52, based on the Scopus criteria for best quartile classification in accounting. It has been published since 2000 by Elsevier Inc, which has its head office in the Netherlands. Therefore, the data cover papers released from 2000 onwards, which allows the assessment of research since the beginning of this century, which, therefore, provides an overview of the last 22 years of research in AIS. Further reasons for selecting this journal include its comparatively higher Scopus ranking among journals that do not include any other terms besides the purposed field of analysis (information systems).

The searching process in Scopus was restricted to those papers with available abstracts for assessment. No additional criteria were applied besides the period and the selected journal. Consequently, no other terms or expressions were used as the key searching term for the papers selected. At the end of this process, 364 papers were gathered from Scopus.

The exploratory analysis performed in this paper provides a breakdown for three distinct periods: from 2000 to 2009, from 2010 to 2019, and, finally, from 2020 to 2022. The first two periods were selected to identify the differences, through a comparative analysis, between the topics from the papers released for the two initial decades of this century. The last, despite being shorter than the previous ones, was particularly proposed to focus the assessment on the trending topics found in the same source.

Textual analysis tools available on NVIVO and {L}exos were used as complementary materials for the main data assessment in this paper. The choice between the two has considered the availability of each tool and the most informative and easily readable information in each context and proposed analysis.

Two main sets of textual analyses are performed, as follows:

(i) The identification of the most frequent terms, which includes the presentation of illustrative word clouds;

(ii) The main set of topics and subtopics from the papers gathered, which includes a complementary diagram of a cluster analysis.

The {L}exos was used to provide the word clouds images underlying the most frequent words (i), while the software NVIVO is used for the analysis of the main set of topics and subtopics (ii).

Regarding the identification of the most frequent terms (i), some words must be previously classified as "stop words" since they have no meaning for research purposes and, consequently, must be removed from the analysis. Since {L}exos has no list of stop

words by default, the NVIVO first suggestion was used as a reference for this purpose. The software provides a set of predefined stop words in different languages (for this case, English was used), which includes the most common prepositions, particles, interjections, unions, adverbs, pronouns, modal verbs, as well as the different forms of common verbs, such as the verbs to be, to have, and similar. The "by default" set of stop words can also be adjusted by users, i.e., included or excluded by users, depending on the purposes of a given analysis.

In this sense, this paper uses the reference list of stop words from NVIVO, as well as frequent "linking words" considered in research papers with no particular meaning for the proposed analysis, such as "paper", "study", "work", "sample", "focus", "aim", "purpose", "contribution", "relevance", "objective", "propose", "study", "effects", impacts", "evidence", "higher", "lower", "significant", and similar ones. Therefore, the complete list of stop words was then included in {L}exos, a freely available website, and two different word clouds from {L}exos were performed by each period under assessment to illustrate the most frequent terms found.

Furthermore, the main set of topics and subtopics, as well as a cluster analysis (ii), were performed using NVIVO. This software uses an automatic process to find the main topics and respective subtopics, as well as the possible relationships (similarities) between them through cluster analysis. The Pearson correlation coefficient is used to identify the similarities, which are indicated by blue lines, with thicker lines highlighting stronger levels.

A review of the papers selected is presented in the following subsection.

### 2.2. Results

This subsection is dedicated to the analysis of the papers published in the *IJAIS* from 2000 to 2022, also considering the three comparative periods proposed. Table 1 presents a summary of the papers selected by period and their textual characteristics, based on information provided by {L}exos. A first synthesis of the textual characteristics of papers by period is therefore provided, including the terms occurring once, the total number of terms, the distinct number of terms and vocabulary density.

**Table 1.** Summary of the textual characteristics of papers selected by period.

| Journal | Period | Number of Papers | Number of Terms Occurring Once | Total Number of Terms | Distinct Number of Terms | Vocabulary Density |
|---|---|---|---|---|---|---|
| *IJAIS* (N = 364) | 2000–2009 | 133 | 1477 | 10,870 | 3045 | 0.28 |
| | 2010–2019 | 174 | 1643 | 13,969 | 3512 | 0.251 |
| | 2020–2022 | 57 | 1059 | 5161 | 1935 | 0.375 |

The figures in Table 1 show a trend for an increasing number regarding the indicators provided, which is also valid for the last three-year period since it already indicates approximately one-third of the figures for the same indicators published in the previous one. The vocabulary density is an exception, however, as the figures decreased from the first to the second period. It is worthwhile to notice that the figures for the distinct number of terms and vocabulary density regarding the last (shortest) period cannot be compared with the previous ones.

Figure 1 presents a summary of the number of papers and the number of citations published in *IJAIS* over the three periods under assessment. The ratio between the number of citations and the number of papers is also provided as a measure of the average impact of papers by period (average citations per period). Notwithstanding, it should be stressed, again, that the discernible distance between the periods under assessment is expected, considering that the most recent periods may have a relative disadvantage when it comes to citations since, usually, older papers have more opportunities to receive citations.

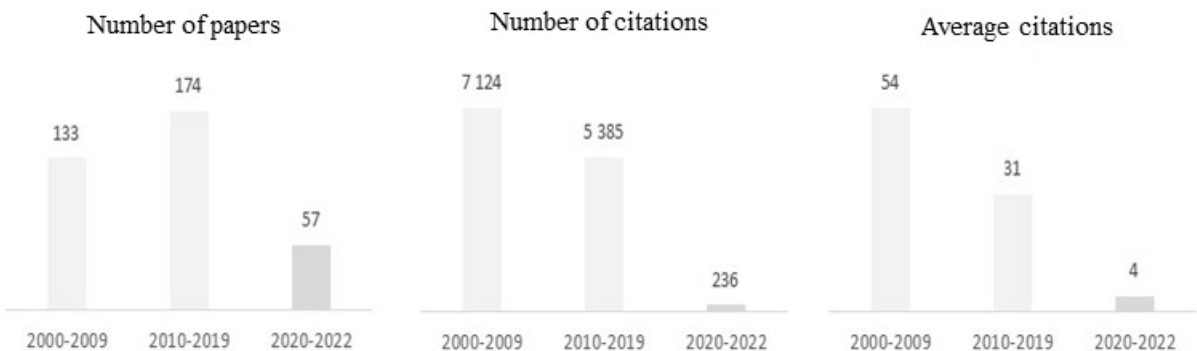

**Figure 1.** Summary of the number of papers and citations by periods in the *IJAIS*.

Figure 1 shows that the number of papers has increased over the period, conversely to the number of citations, which tends to be more noticeable for the last period, as expected.

Among the 25 papers with more than 100 citations in *IJAIS*, only 7 were published in the second period of analysis, with the most cited paper reaching 202 citations [16]. This is followed by Lee et al. [17], which has 194 citations, with the remaining reaching a maximum of 130 citations [18]. These papers include diverse themes, including the use of dashboards to assess performance management, partial least squares path modelling in accounting research, or business analytics and ERP in management accounting. On the other hand, the most cited paper for the first period [19], on ERP implementation, had a maximum of 437 citations, which is followed by two papers with more than 300 citations [20,21]. The oldest is also focused on ERP but from the perspective of its financial impacts, whereas the most recent covers business intelligence as a topic of emerging technologies.

The findings are presented in the next subsubsections, with a breakdown by period.

2.2.1. Results for the First Period (2000 to 2009)

A recurrent theme during the first period (2000 to 2009) was the use and implementation of AIS, in general, and enterprise resource planning (ERP) including papers developed as literature reviews (e.g., [19,20,22–32]).

Furthermore, studies on IT governance practices and models, in general, and the resource-event-agent (REA) model can also be identified (e.g., [33–35]). There are also analyses related to performance, integrity, risks, success factors, obstacles, or challenges associated with implementing technologies in different areas of accounting that relate to the previous themes, some of them discussing the advantages and disadvantages of outsourcing (e.g., [36–42]).

Other topics emerged or gained prominence during the period, namely the analyses related to the following topics:

i.    The dissemination of information via the Internet, such as web reporting, e-commerce models, or web-based business or services (e.g., [43–48]);

ii.    The use, implementation, and implications of continuous auditing or continuous monitoring (for instance, [49,50]);

iii.    The use or implementation of languages or taxonomies based on Extensible Markup Language (XML) or Extensible Business Reporting Language (XBRL), sometimes also linked to previous themes (e.g., [49,51–53]);

iv.    The implementation of internal control system models in the IT area, such as control objectives for information and related technology (COBIT), the relevance of IT certification, such as WebTrust, and also the topics from the Information Systems Audit and Control Association (ISACA) curricula (for instance, [54–56]);

v.    Finally, although less incipient in this period, some studies on the emerging technologies from the beginning of this century, such as neural networks (e.g., [57]), or business intelligence (e.g., [21]).

Following, Figure 2 provides the word clouds for this period to illustrate the most common words.

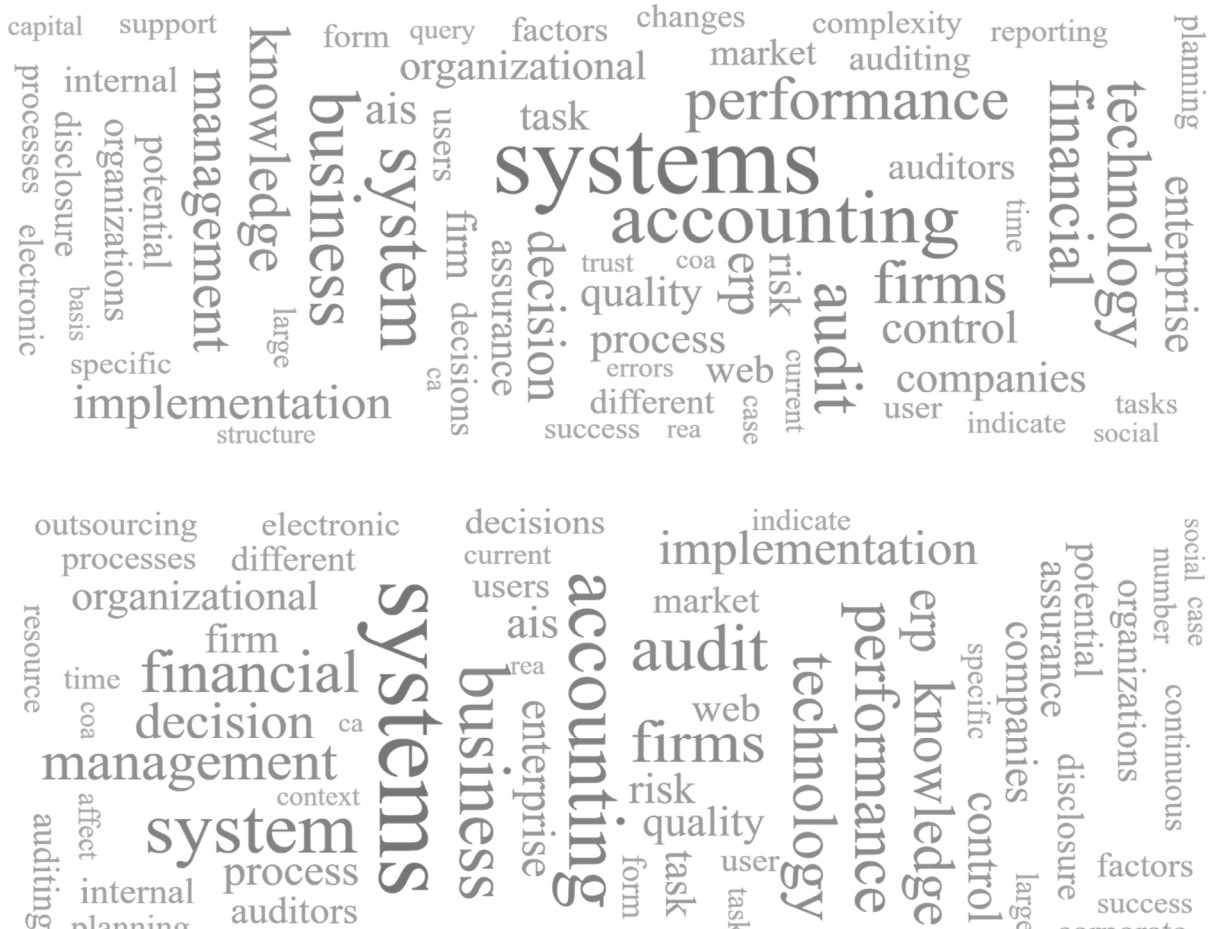

**Figure 2.** Word clouds from the papers in the *IJAIS* (from 2000 to 2009).

From Figure 2, terms such as "accounting", "information", and "systems", as well as "companies", "organization", "firm", and, "enterprise", are associated with others such as "implementing", "process", "design" or "development", as well as "effectiveness", "performance", "management", "knowledge", "organizational", which indicate the prevalence of studies mostly dedicated to AIS and ERP, also included in the list.

The previous terms along with others such as "audit", "auditors" or "auditing", "risk", "assurance" and "continuous" probably highlight studies related to continuous auditing. Lastly, the terms "disclosure", "electronic" and "corporate" may refer to studies on continuous reporting, e-reporting and corporate websites as a new channel for information disclosures by companies. Conversely, topics such as XML, XBRL, internal control models in IT, or the use or implementation of emerging technologies appear to be less prominent in this first period.

The terms "web" and "trust" also evidence the inclusion of this topic within the papers published in this first period of analysis in the *IJAIS*.

Therefore, to strengthen the previous indications, a global prominence of research on "AIS", and the "ERP", as well as those related to them, can be found for this period, regardless of the exceptions provided above, mostly related to the processes associated with the audit and reporting areas in general, including issues on risk, security, and internet developments.

Figure 3 provides the topics automatically codified as the most relevant from the textual analysis for this first period.

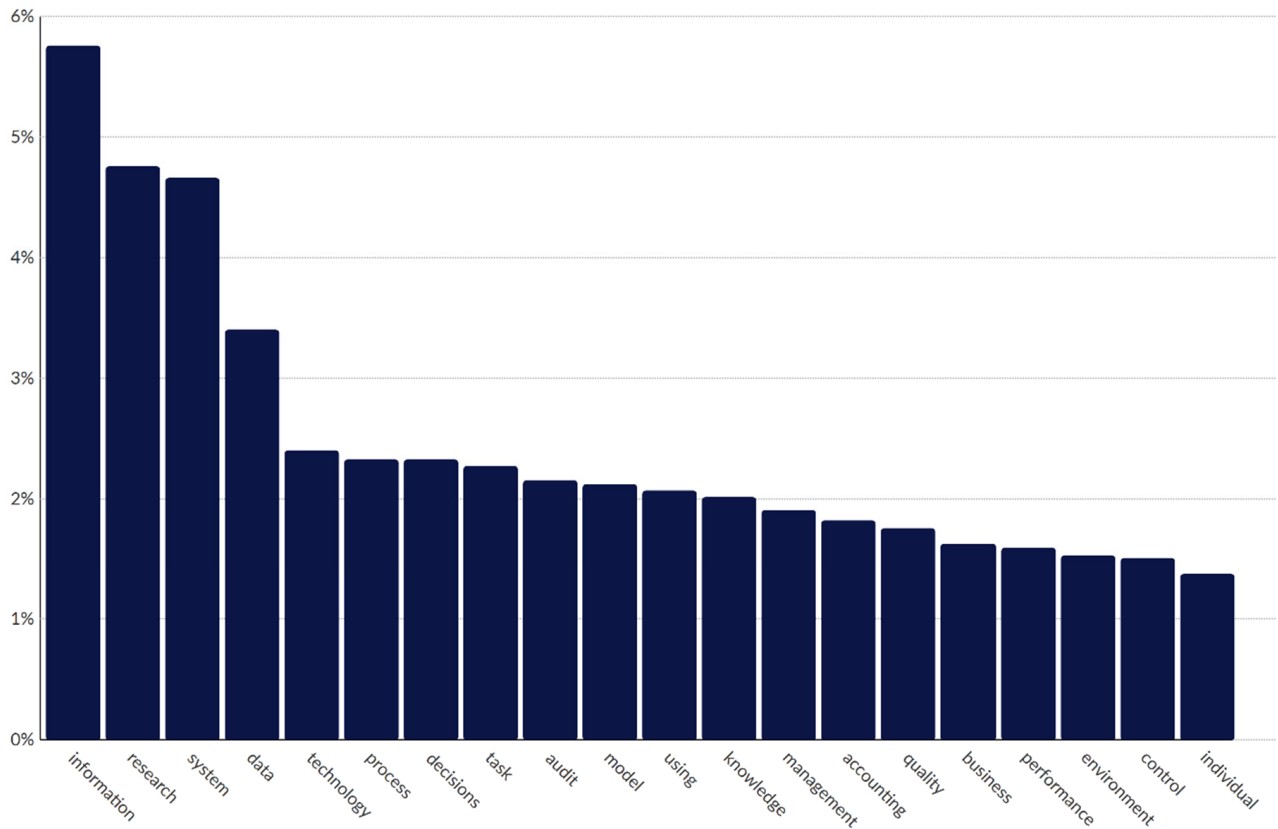

**Figure 3.** Topics automatically found from the papers in the *IJAIS* (from 2000 to 2009).

Figure 3 shows the 20 most relevant topics that were automatically codified according to their frequencies in a different set of expressions found in papers from this period. The topics "information", "research", "system" and "data" are the most prominent, with the first three having more than 4% of relative importance. Other relevant topics not listed in Figure 3, despite being identified as relevant topics, include "design", "effectiveness" and "structure". Below, the words highlighted in bold identify those topics which are commonly shared (cross-references) among those 20 most relevant, as a useful procedure to identify the focus of researchers in this period:

1.  **Accounting**: accounting academics, accounting data points, accounting informa-tion, **accounting knowledge, accounting research productivity, accounting system design, accounting systems research, accounting tasks**, database accounting, **differentiating accounting systems**, financial accounting literature, leading accounting, managerial accounting perspective, **quality accounting publication**;

2.  **Audit**: audit automation constructs, audit committees, audit documentation, audit engagement risk, audit opinion, audit trail, audit work, auditing education, auditing literature, computer-assisted auditing techniques, continuous audit, **current audit environment**, specific auditing concerns, various audit domains;

3.  **Business**: business clients, **business information**, business operations, business or-ganizations, **business process diagrams**, **business process level**, **business process modelling conventions**, business world, Canadian business units, everyday business communications, extensible business, strategic business planning, web-based business;

4.  **Control**: control group, control objectives, control relationships, **designing control systems**, external controls, **hierarchical control structures**, informal controls, inter-nal control, international control guideline, proper control procedures, using con-trol charts;

5. **Data**: **accounting data points**, **data quality**, data streams, data warehouses, electronic data interchange, financial data, including data flow diagrams, **normal form data structure**, numerical data increases, secondary data analysis, site data, specific data, spending data, underlying data trends, **unnormalized data structure**, using data;

6. **Decisions**: bonus allocation decisions, **decision aid research**, decision aid use, decision aids, decision facilitation, **decision process**, investment decisions, **management decision models**, multiple decision, novice decision makers, operational decisions, repetitive choice decisions, repetitive valuation decisions, user decisions;

7. **Environment**: alternative environments, continuous reporting environment, **current audit environment,** external environment, manufacturing environments, traditional reporting environment, virtual team environment;

8. **Individual**: individual characteristics, individual decision-makers, individual determinants, individual faculty members, individual faculty productivity, individual level, individual provider attributes, individual units, judging individuals, perceiving individuals;

9. **Information**: **accounting information**, **advanced information technology**, **business information**, **computer-supported information systems**, corporate reporting information, decision-making information, emerging information needs, emerging information technologies, financial reporting information, financial statement information, future-oriented information, **human information processing, important information processing mechanism**, information age, information content, information integrity attributes, information load, information location, information requests, information security, **information system designs**, **information systems research**, informationally equivalent, inter-organizational information sharing, **low information quality seal**, **management information systems**, **management information value chain**, nonfinancial information, online information, open information sharing, output information, preliminary information, **specific information technology**, **supporting information technology**, varied information, vast information source;

10. **Knowledge**: **accounting knowledge**, additional knowledge, causing knowledge acquisition, **expert-like knowledge structures**, feedback impacts knowledge acquisition, filtering knowledge, improving knowledge workers, **knowledge management focus**, **knowledge management practices**, **knowledge management system**, procedural knowledge;

11. **Management**: **cost management systems, effective management**, hybrid manager profile, impression management, **knowledge management focus, knowledge management practices, knowledge management system, management decision models, management information systems, management information value chain**, senior managers, top management support;

12. **Model**: **business process modelling conventions**, conceptual model, contingency model, enterprise modelling, er model, **management decision models**, mathematical model, **research model**, residual income valuation model, **task circumplex model**, theoretical model;

13. **Performance**: firm performance, **managerial end-user performance**, organisational performance, organizational performance, performance evaluation, performance outcomes, subsequent decision-making performance, superior performance, **task performance**, traditional performance measures;

14. **Process**: assurance process, **business process diagrams, business process level, business process modeling conventions**, decision process, event process chains, extensive sample selection process, **human information processing, important information processing mechanism**, little processing, process level risk assessment, processing view, production processes, stable processes, standard development process;

15. **Quality**: **data quality**, disseminating quality, **low information quality seal**, **quality accounting publication**, quality measurement component, quality outlets, quality perspective, service quality, **system quality**;

16. **Research**: **accounting research productivity**, **accounting systems research**, additional research, ais research, attitudinal ambivalence research, case research, current research work, decision aid research, development research, empirical research, field research, future research, **information systems research**, little research, past research, previous research, prior research, recent research, research community, research domain, research findings, research hypotheses, research instrument, research issues, research method opportunities, **research model**, research propositions, **research prototype system**, research questions, research survey, rich research opportunities, various research methods;

17. **System: accounting system design, accounting systems research**, alternative measurement systems, automated record system, **computer-supported information systems, cost management systems**, current system, **designing control systems, differentiating accounting systems, electronic audit-work paper system**, enterprise systems implementation, **expert system groups, expert system types, expert system users, information system designs, information systems research,** key system components, **knowledge management system, management information systems**, medical record system, **research prototype system**, successful system, system acceptance, system acquisition, **system design alternatives, system effectiveness**, system implementation changes, system integration, system outputs, **system quality**, system transformation, system usage, **systems design** scenarios, tertiary assurance system, work systems;

18. **Task**: **accounting tasks**, brainstorming tasks, complex tasks, decision-making tasks, financial tasks, optimisation tasks, querying tasks, simple tasks, **specific design tasks**, task accuracy, task characteristics, **task circumplex model**, task completion, task force, **task performance**, task requirements, task routineness;

19. Technology: advanced information technology, emerging information technologies, emerging technologies, learning technology, specific information technology, supporting information technology, technological advances, technological determinism, technological discourse, technology complexities, technology features, technology fit, technology medium;

20. **Use: auditor use**, continued use, decision aid use, **expert system users**, user acceptance, **user decisions**, user requests, user satisfaction, using activity, **using control charts**, **using data**, using eighty-nine, using paper methods.

The results suggest the prominence of research in AIS dedicated to improving the information (e.g., through its design, quality, structure, and easier visualisation) for management (internal) purposes (e.g., knowledge, effectiveness, and support for decision-makers), which is the traditional investigation line in this area. On the other hand, the topic of "individual" (the 20th one) was the exclusive case for which no relationships were found. Furthermore, additional topics also deserve references within the different topics, such as continuous audit, continuous reporting, web-based businesses, virtual teams, and extensible business (e.g., XBRL). Finally, emerging technologies can also be found as a general reference but with no significant level of specific pieces of evidence regarding topics currently associated with this concept.

Finally, Figure 4 presents the cluster analysis performed to illustrate the relationship between the topics and subtopics for this first period.

The cluster analysis shown in Figure 4 indicates the link between AIS and data management for businesses and control purposes (i.e., information for the decision-making process), which underlies the papers in this period, as highlighted in the aforementioned analysis. Moreover, the emergence of continuous auditing and reporting in this period is also evident.

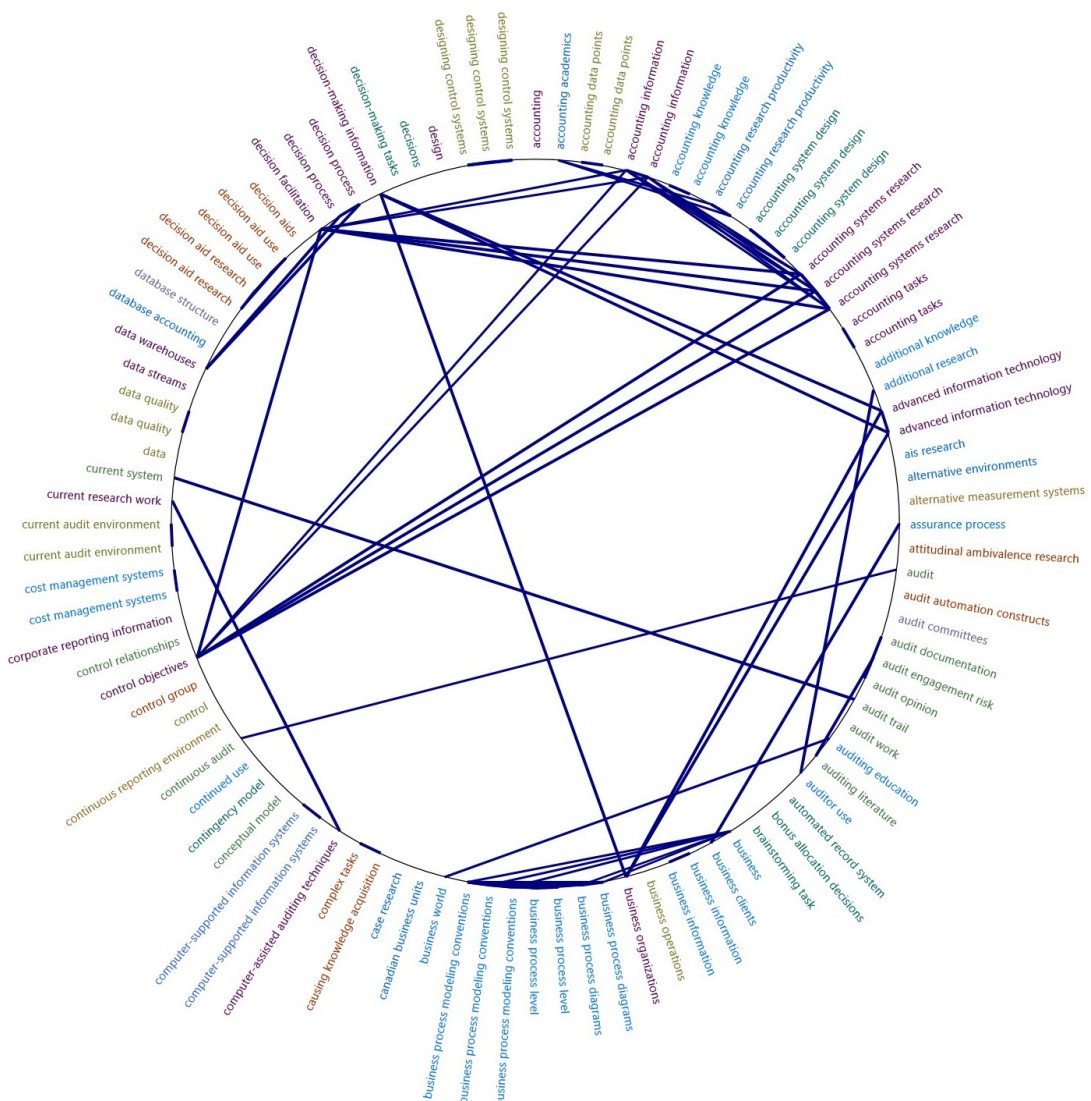

**Figure 4.** Cluster analysis for the papers in the *IJAIS* (from 2000 to 2009).

### 2.2.2. Results for the Second Period (2010 to 2019)

Regarding the second period of analysis (2010 to 2019), it can also be observed that AIS and ERP remain relevant topics (e.g., [58–67]), despite sharing their relative importance. The AIS has been the subject of a significant number of studies developed during this period, including additional literature reviews and critical perspectives. Likewise, one of the most cited papers, presenting the same methodological perspective, is the only study identified on dashboard use [16]. Notwithstanding, studies in the following areas gained greater prominence over those initiated in the previous period:

i. Continuous auditing or continuous monitoring (e.g., [68–72]);
ii. Languages or taxonomies based on XML (less expressive) or XBRL, in a more expressive number than those found in the previous period (e.g., [73–80]);
iii. Business intelligence or business analytics (mostly) (e.g., [19,81–83]);
iv. Artificial intelligence, data analytics, big data treatment, and the use of machine learning and data mining techniques (mostly), which are sometimes associated with previous themes and, in many cases, dedicated to the definition of processes related to the detection of fraud, misreporting or tax evasion (e.g., [84–92]);
v. Cloud computing (e.g., [67,93–95]);
vi. Blockchain-based technologies, which are mostly evidenced at the end of this decade (for instance, [96,97]).

In addition to the areas most linked to emerging technologies, studies are identified in more generic areas of process management or IT infrastructure, in terms of processes, governance, associated risks, and value added, among others, also including studies on the use of outsourcing or, later in this decade, on cybersecurity issues (e.g., [98–106]). Moreover, topics such as e-commerce or the dissemination of financial information over the Internet, although existing, have less profusion in this period (e.g., [107]). The first studies on the use of social media and online communities, on the other hand, appeared as an emerging topic in this period (e.g., [108,109]). It is also from this period, albeit residually, studies based on nonfinancial information as a source of data, such as sustainability reports or integrated reporting (e.g., [110]), appeared.

Figure 5 presents the word clouds to illustrate the most common words for this second period.

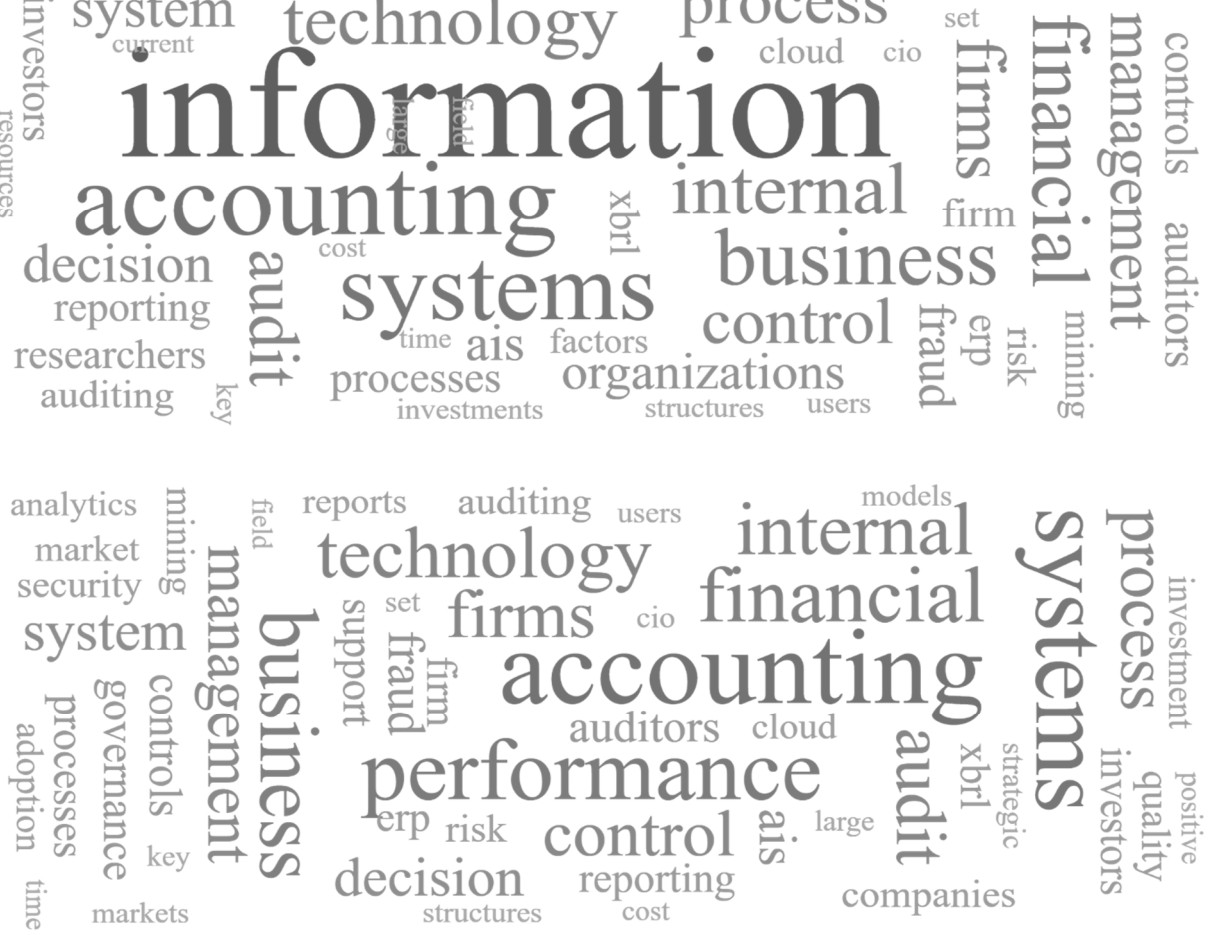

**Figure 5.** Word clouds from the papers in the *IJAIS* (from 2010 to 2019).

Figure 5 shows new terms usually classified as emerging technologies, such as "XBRL", "mining" and "cloud". In addition, studies related to fraud detection, sometimes associated with the use of data mining-based technologies, also play an important role, as evidenced by the identification of the term "fraud". On the other hand, the issues of continuous auditing and outsourcing of resources do not seem to be among the most important compared to the previous period, although they are also present in this decade.

Finally, in the context of the more traditional issues, such as topics related to business and IT architecture or computer security, also emerge, as evidenced by the greatest relevance of terms such as "business", "process" and "control", as well as the emergence of new terms in this set, such as "governance" and "security".

Following, Figure 6 presents the topics automatically codified as the most relevant from the textual analysis for this second period.

**Figure 6.** Topics automatically found from the papers in the *IJAIS* (from 2010 to 2019).

Based on the same procedure performed before, Figure 6 shows the 20 most relevant topics. In comparison to the previous period, and despite commonly sharing the three most relevant topics, none of them reached the 4% of relative importance found earlier. Some topics emerge as novelties in this period, such as "reporting", "disclosure", "measurement", "firm" and "function". The set of topics is significantly higher and diversified in this second period, totaling 55 concepts, which include, for instance, "data mining", "experimental", "fraud", "governance", "investors", "market", "risk", "security", "standards", "statements", "trading", and "value". This is also evident in Figure 6 by the set of topics for which the relative importance found is lower than 2% for this period (12 versus 8 for the previous period). Then, the 20 most relevant and the relationships between them and of the automatic topics found, highlighted in bold, are presented below:

1.  **Accounting**: 129 accounting students, accounting benefits, accounting domains, **accounting information systems academics**, **accounting information systems field**, accounting journals, accounting literature, **accounting processes**, accounting publications, **accounting researchers**, **accounting standards**, annual bank account balances, especially management accounting, **firm accounting performance, outsourcing accounting functions**;

2.  **Audit**: audit analytics, audit arena, **audit fraud brainstorming**, **audit process**, **audit standards**, **audit support systems**, audit team, auditing literature, budgeted audit hours, chief audit executives, computer audit specialist, continuous auditing methodology, **current audit practice**, financial audits, **internal audit function**, internal auditing department, prior year audit, **reduced audit fee increases, small audit firms, traditional audit paradigm**;

3.  **Business**: business networks, business operations, **business process agility, business process standards**, **business value research**, computing-related business objectives, **existing business processes**, extensible business, hindering business efforts, **in-**

**termediate business processes**, **overall business performance**, **reporting business information**;

4. **Control**: control compliance, control issues, corrective controls, effective controls, ineffective controls, **informal management control systems**, internal control deficiencies, internal control environment, internal control overrides, **internal control reporting requirements**, internal control weakness, internal control weaknesses, it-related controls;

5. Data: applying data mining techniques, corporate data, data analysis tool, data patterns, descriptive data mining approach, descriptive data mining strategy, financial data, global data ecosystem, journal entry data sets, out-of-sample data, panel data, perceptive field survey data, precise data values, prediction data mining techniques, process-level data, procurement data, proprietary data, quantitative data, researching journal entry data mining, semi-monthly data, soft copy data, tagged data, using data;

6. **Decisions**: compared decisions, deception detection decision aid, decision aid reliability, decision aid reliance behavior, decision aids, decision problems, **decision processes**, decision trees, **experimental decision aid research spans**, **governance decision making**, optimal decision, outsourcing decision, reliance decision;

7. **Disclosure**: **cybersecurity disclosure guidance**, **cybersecurity risk disclosure**, disclosure credibility, disclosure role, environmental disclosures, extensive disclosure, **financial statement disclosures**, improving disclosure timeliness, issuing video disclosures, unauthorized disclosure;

8. **Effective**: brainstorming effectiveness, compromising regulation effectiveness, detrimental effect, differential effect, **effective controls**, halo effect, **information environment effects**, interactive effect, mean effects, positive effect, profound effect;

9. **Firm**: aggregate firm level, appointing firms, durable goods industry firms, **firm accounting performance**, firm productivity, firm profitability, **firm value**, firm years, registered firms, **small audit firms**, **superior firm performance**, threatened firms;

10. **Function**: bi-planning functionality, **bi-reporting functionality**, incompatible functions, **internal audit function**, outsourcing accounting functions;

11. **Information: accounting information systems academics, accounting information systems field, bank trading information systems**, capturing context information, chief information officer, deceptive information, detail-tagged footnote information, **financial reporting information, health information technology expenses**, information asymmetry, information environment effects, **information quality**, information release, **information security risk management, information systems professionals, information systems researchers, information technology literature, information technology outsourcing**, integrating information, **performance measurement information**, qualitative information, **recent high-profile information security breach incidents, reporting business information, risk information increases**, supplemental information displays, ™ **information processing costs, user satisfaction measure information system success, using information**;

12. **Management**: bank management, cloud management committee, entail managers, **environmental management approach, especially management accounting, informal management control systems, information security risk management**, management assertions, management support, managing expectations, resource management;

13. **Measures**: measuring spreadsheet infusion, perceptual measures, **performance measurement capabilities, performance measurement information, quality measures**, quantitative measure, **strategic performance measurement system**, subjective measures, **user satisfaction measure information system success**;

14. **Performance**: average performance, **firm accounting performance, firm-level performance**, future performance goals, **internal process-level performance**, organizational performance, **overall business performance, performance measurement capabilities, performance measurement information, strategic performance measurement system, superior firm performance**, supply chain performance;

15. **Process: accounting processes,** assurance process, **audit process, business process agility, business process standards**, close process, **decision processes**, estimation process, **existing business processes**, implementation processes, **intermediate business processes**, labor process, manual process, natural language processing, order fulfilment processes, process efficiency, **process level,** strategic erm processes, ™ **information processing costs**, work processes;

16. **Reporting**: annual reports, digital reporting, discretionary reporting, **financial reporting information, financial reporting systems, internal control reporting requirements**, internet reporting, **reporting business information**, reporting language, reporting timeliness, required reporting deadlines, **standard reports, traditional business-to-government reporting**;

17. **Research**: artificial intelligence research, broad research streams, **business value research**, collaborative design research, **experimental decision aid research spans, expert systems research**, future research, multi-method research, potential research, prior research, research discipline, research environments, research methodology, research perspectives, research program, **research quality**, **research settings**, **researching journal entry data mining**, right research, **traditional research classification**;

18. **System**: 43 expert systems, **accounting information systems academics, accounting information systems field**, accounting-related expert systems papers, **audit support systems**, automated systems, **bank trading information systems**, computer-mediated communication system, **decision-aid system**, enterprise resource planning systems adoption, enterprise systems results, **expert system publications, expert systems research, financial reporting systems, incentive systems, informal management control systems, information systems professionals, information systems researchers**, manual systems, restrictive systems, **strategic performance measurement system, system quality**, transparent system, **user satisfaction measure information system success**;

19. **Technological: health information technology expenses, information technology literature, information technology outsourcing**, supporting technologies, **technological competence**, technological domain, technological solutions, technology dominance;

20. **Using**: emergent use, managerial use, media use, **practice use**, spreadsheet use, **user satisfaction measure information system success, using activity theory, using data**, using a hospital, **using information**, using responses.

The results above stressed the issues related to reporting and disclosure, particularly those prepared under audit and financial standards. Furthermore, governance matters, risk and cybersecurity, as well as different perspectives of performance assessments, are topics that can be easily found in the pieces of evidence above. Finally, conversely to the previous period, emerging technologies can be more easily exemplified from the references found in the topics above through the terms such as artificial intelligence and data mining.

Finally, Figure 7 presents the cluster analysis performed to illustrate the relationship between the topics and subtopics for this second period.

When compared to the same analysis performed for the previous period, Figure 7 indicates a lower level of relationships between the topics and subtopics underlying the papers in this period. Moreover, emerging technologies such as artificial intelligence and data mining, as well as a more diversified set of topics on auditing and information systems in general, are also highlighted.

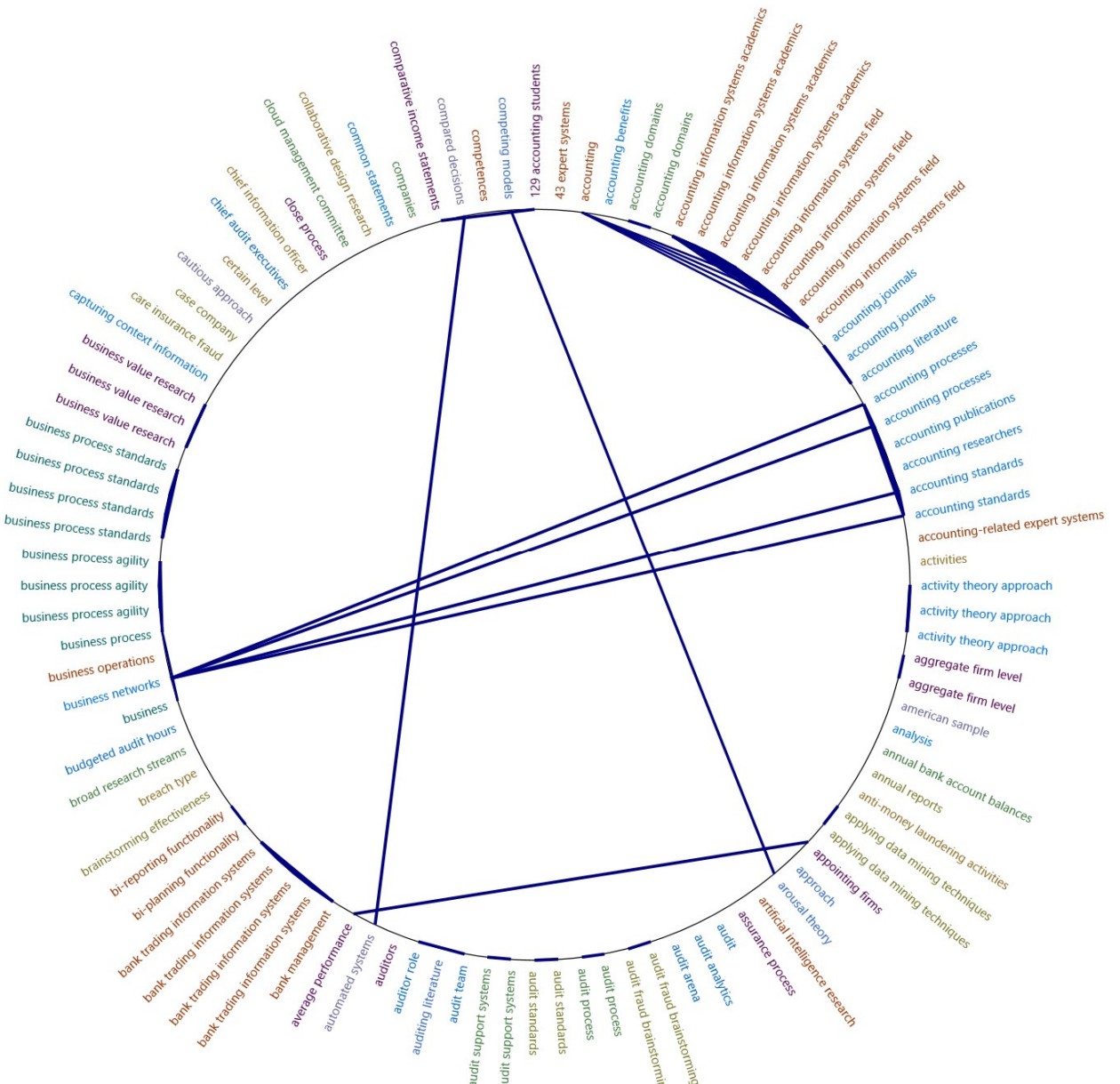

**Figure 7.** Cluster analysis for the papers in the *IJAIS* (from 2010 to 2019).

### 2.2.3. Results for the Latest Period (2020 to 2022)

Regarding the last period of publications in the *IJAIS*, studies on the so-called emerging technologies that started at the end of the last decade have followed. Furthermore, it can be seen a lower development of the traditional topics from the earlier periods, such as the use of AIS and ERP (e.g., [111–113]) or the control and management of IT infrastructures (e.g., [114,115]). Those newest studies seem to offer a wider variety of perspectives that address issues such as blockchain-based technologies (e.g., [116–119]), which include crypto-assets or cybersecurity issues (e.g., [120–125]). Less numerous, though available, are also studies on social media (e.g., [126,127]).

As for the topics of emerging technologies consolidated in the last decade, there is still opportunity for the various topics that have been previously explored, such as continuous auditing or continuous monitoring, data analytics, big data, machine learning, artificial intelligence, and cloud computing (e.g., [128–137]).

Although not new, studies can also be found that link different topics in a single research, namely the study by Zhang et al. [137], which addresses the issue of continuous

monitoring using machine learning and interactive visualisation of data. Another example is the research by Alzamil et al. [138], which uses text-mining tools to assess financial information published on social networks. Finally, it is important to notice the existence of a single study on the COVID-19 pandemic [139] and the development observed in studies that use nonfinancial information as a source of information, such as sustainability reports or integrated reporting (e.g., [130,140]).

Figure 8 provides the word clouds to illustrate the most common words for this last period.

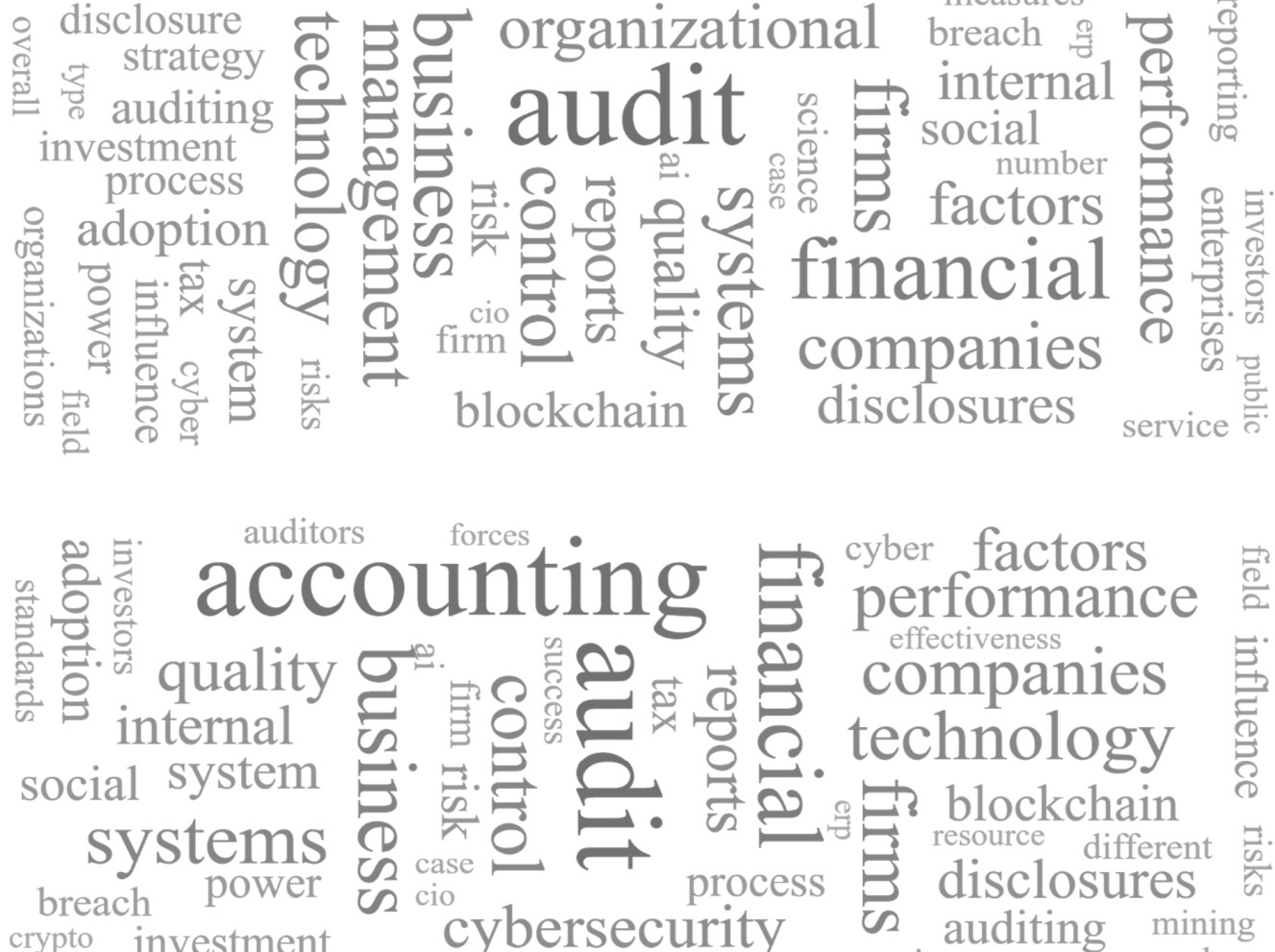

**Figure 8.** Word clouds from the papers in the *IJAIS* (from 2020 to 2022).

Furthermore, "cybersecurity", "breach" and "blockchain" arise among the most common terms found in this period, replacing topics such as "XBRL" as emerging topics from the previous one. On the other hand, differences in the relative frequencies of terms such as "control" and "quality", together with the evidence of traditional ones, such as "reports", "reporting" and "decision-making", also indicate new approaches for the research developed in this period, in comparison to the previous ones.

Therefore, this latest three-year period seems to have a more diversified range of topics on emerging technologies, methods, and approaches in accounting research.

Following, Figure 9 presents the topics automatically codified as the most relevant from the textual analysis for this last period.

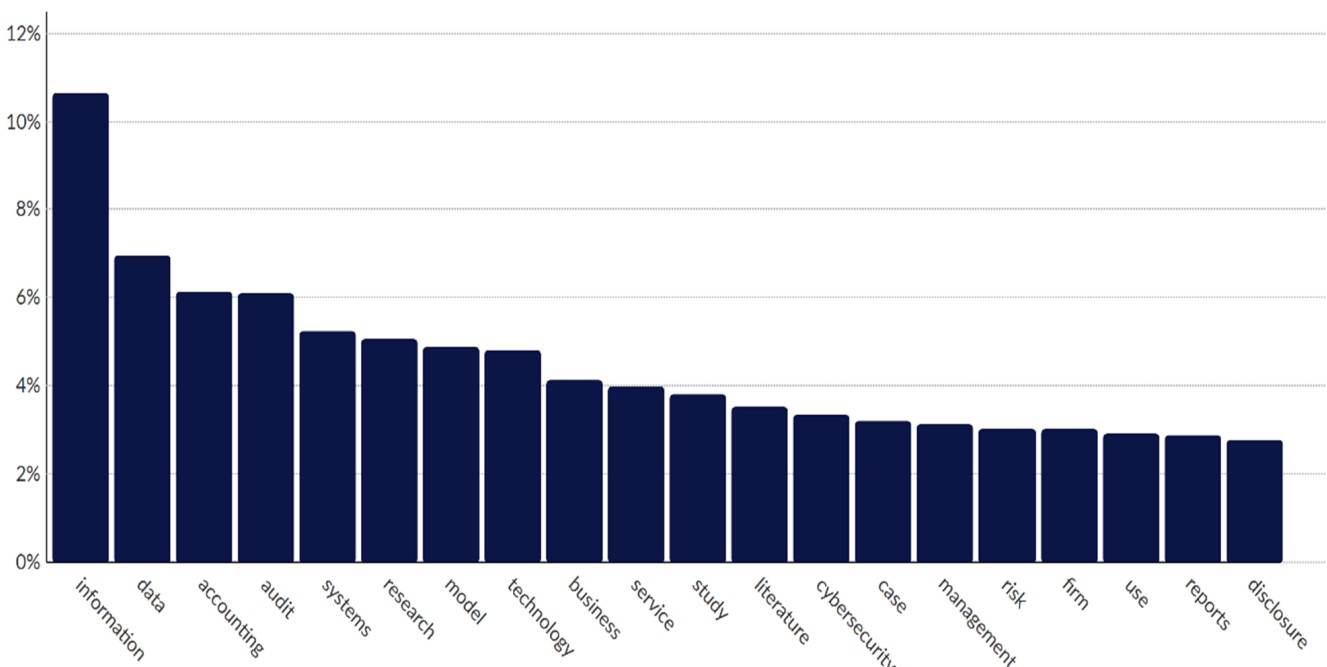

**Figure 9.** Topics automatically found from the papers in the *IJAIS* (from 2020 to 2022).

Figure 9 shows that the gap between the topic "information", which is kept as the most relevant, is amplified in this last period. Despite being the shortest period under assessment, 31 topics were automatically identified, including, for instance, "blockchain", "effects", "factors", "internal control", "material weakness", "process", "quality", "tax", and "theory", besides those found in Figure 8. Below, the 20 most relevant, highlighted in bold, are listed, based on the same procedures as used before:

1.  **Accounting**: 136 accounting professionals, accounting context, **accounting data, accounting fraud data mining literature, accounting fraud detection models, accounting information systems case study, accounting information systems scholars, accounting literature**, aggregated accounting numbers, cloud-based client accounting, **developing accounting information systems**, different accounting standards, highest-ranked accounting journals, laggard accounting systems, **management accounting**, professional accounting bodies, recent accounting fraud theory, robust account;

2.  **Audit**: **4 audit firms**, asset-related audits, audit conclusions, audit fee premiums, audit framework, audit hours, audit personnel, audit practice, audited entity, auditing parties, auditing profession, computer-assisted audit tools, contemporary audit standards, continuous audit procedures, **cybersecurity audit effectiveness**, digital audit evidence, financial statement audits, **improving audit quality**, increasing audit productivity, **internal control audit work**, manual audit procedures, recurring audit deficiencies, relevant audit standards, reliable audit evidence;

3.  **Business**: business digitization, **business information technology intensity**, business model transformation indices, business operations, **business processes**, business professionals, business rules, general business descriptions, increasing business competition, strategic business partner role;

4.  **Case**: accounting information systems case study, compelling use case, in-depth case study, participatory case study, specific case, various use cases;

5.  **Cybersecurity**: 52 cybersecurity comment letters, **cybersecurity audit effectiveness**, cybersecurity breach incidents, cybersecurity incidents, **cybersecurity risk disclosure practices, cybersecurity risk disclosure trends**, organizational cybersecurity risk exposure, overall cybersecurity risks, **proprietary cybersecurity information**, regarding cybersecurity;

6. **Data**: **accounting data**, **accounting fraud data mining literature**, available data, big data capabilities, **big data technologies**, climate data, collected data, data analytics, data breach, **data processing integrity**, **data quality research**, data standards, data visualization software, financial data, general ledger data, interactive data visualization, interview data, legal-entity data segmentation, novel data analysis technique, novel data mining technique, real-life data, social media data, textual data, unstructured data, **using data**;

7. **Disclosures**: corporate disclosures, cryptocurrency disclosures, **cybersecurity risk disclosure practices**, **cybersecurity risk disclosure trends**, disclosure location, **firm disclosures**, remediation disclosures;

8. **Firms**: **4 audit firms**, adopting firm, **firm disclosures**, firm resource, firm samples, firm size, firm tenure, incentivizing client firms, Korean-listed firms;

9. **Information**: accounting information systems, accounting information systems case study, accounting information systems scholars, agricultural information systems, budget information, **business information technology intensity**, clarifying information, **developing accounting information systems**, existing information systems, **information content, information dissemination, information overload, information processing capabilities, information quality, information systems discipline, information systems theories, information technology experts, integrated information systems, personal information management capabilities**, private information, **proprietary cybersecurity information**, qualitative information, quantitative information, social responsibility information, specific value information, text information;

10. **Literature**: **accounting fraud data mining literature**, **accounting literature**, current literature, existing literature, extensive literature review, **natural language processing literature**, prior literature, research literature, systematic literature review;

11. **Management**: **cyber risk management effectiveness, cyber risk management maturity, knowledge management research, management accountants, management accounting, management reporting**, personal information management capabilities, supply chain management, top management commitment, top management support, workflow management;

12. **Model**: **accounting fraud detection models**, **business model transformation indices**, developed models, digital maturity model, filed model, **force field model**, model performance, predictive models, proposed model, research model, theoretical model, **wave theory life cycle model**;

13. **Reporting**: financial reporting, **management reporting**, report length, social responsibility reports, unique reporting requirements;

14. **Research**: answering research questions, **data quality research**, design science research contribution, empirical research, extending research, future research, **knowledge management research**, prior research, recent research, research initiative, **research literature**, **research model**, research studies;

15. **Risk**: cyber risk management effectiveness, cyber risk management maturity, cybersecurity risk disclosure practices, cybersecurity risk disclosure trends, organizational cybersecurity risk exposure, overall cybersecurity risks, regulation risks, risk assessment;

16. **Service**: assurance services, consumer services, payment services, service components, **service quality**, shared service mode;

17. **Study**: **accounting information systems case study**, cross-sectional field study, existing studies, **in-depth case study**, longitudinal study, **multi-case study approach, participatory case study, research studies**;

18. **System**: accounting information systems, accounting information systems case study, accounting information systems scholars, agricultural information systems, developing accounting information systems, enterprise resource planning system design agenda, existing information systems, information systems discipline, in-

formation systems theories, integrated information systems, intelligent systems, laggard accounting systems, system quality, system** usage;

19.  **Technology**: above-mentioned technologies, **big data technologies, blockchain technology applications, blockchain technology solutions, business information technology intensity**, computer technology, emerging technology adoption, **information technology experts**, learning technologies, ledger technology, past technology experience, technological advancements, technological developments, trending technology;

20.  **Use**: **compelling use case**, **cost-effective use**, decision-making use, **effective use**, organizational use, user satisfaction, **using data**, using DevOps, using propensity score, **various use cases**.

From the most relevant topics or subtopics found for this period, distinctive characteristics can be highlighted, as follows:

- A more diversified set of subtopics within the topics found;
- The increasing relevance of matters regarding social responsibility, climate, and budgetary information;
- A more evident link, from the subtopics, identified, between accounting and other social sciences through the consideration or inclusion of a wide-ranging of explanatory factors, such as "individual factors", "environmental factors", "organizational complexity factors", "psychological factors", and "social-psychological factors";
- A diverse set of underlying theories and research methods used, particularly those focusing on case and experimental studies, as well as literature reviews (for this reason, "case", "study", and "literature" appears as novelties within the most relevant topics, besides those previously found, such as "research" and "model");
- Besides general references to "emergent technologies" or "trending technologies", the most significant number of indications on specific uses of those tools as precise topics or subtopics, for instance, "blockchain", "crypto assets" or "cryptocurrencies", "intelligent systems", "cloud-based accounting", "big data", "data analytics", "data mining, "learning technologies", and "natural language";
- "Risk", "cybersecurity" and "tax" are included in the set of main topics, which demonstrates the growing relevance of those topics (the latter as a particular novelty for this period).

Finally, Figure 10 presents the cluster analysis performed to illustrate the relationship between the topics or subtopics for this latest period.

Figure 10 shows a most diversified set of topics or subtopics and relationships, as well as a particular increase in case studies and literature reviews, which may indicate the prominence of those research methods in this latest period, as discussed before. Regardless of the relevance of the traditional topics in AIS research, the figure also highlights the relevance of usual topics included in the concept of emerging technologies, such as big data, cloud computing, cryptocurrencies and blockchain, added to those found in previous periods (for instance, data mining, continuous audit and cybersecurity).

The following section discusses the main findings, presenting the main limitations and proposing prospects for further research in accounting.

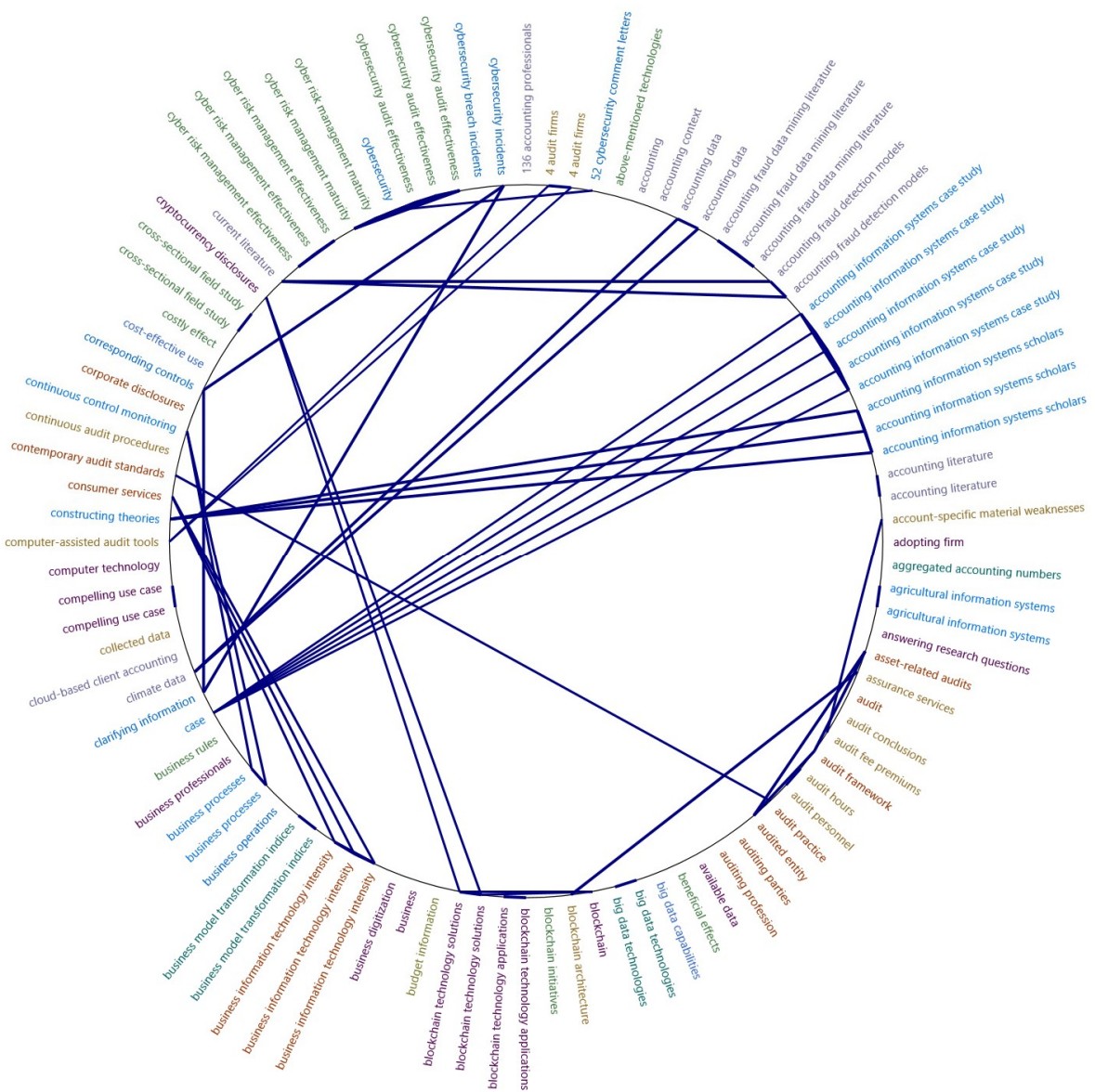

**Figure 10.** Cluster analysis for the papers in the *IJAIS* (from 2020 to 2022).

### 3. Discussion

This paper selected one top journal in accounting, which is specifically dedicated to the use of technologies in accounting, to assess the research topics published and their evolution over this century (from 2000 to 2022). For this purpose, the analysis compares three distinct periods, namely the first two decades and the last three years. The latest period is particularly used as a proxy for the recent trends in accounting research.

The findings indicate that trending topics in accounting have increased their relevance and influence, enriching the debate and future perspectives for professionals, academics, and researchers in this area through the combined use of new technologies. This aspect is also stressed by Kroon et al. [14], who concluded that the assessment of their impacts on the accounting profession is scarce, suggesting that future research should improve the empirical analysis of their contributions to open innovation.

In this context, it is worthwhile to mention that professionals, academics, and researchers have highlighted the difficulty of adequacy and monitoring the developments in the accounting profession and education due to new topics and technological advancements. Authors argue that accounting research needs to evolve and follow contemporary

trends in reporting, regulation, work methods, social and political trends, as well as the technological environment, with a stronger connection between professionals, academia, and researchers (e.g., [1–3]). Therefore, following such challenges, and keeping up with their needs and demands are relevant matters for different subjects. Collaborations between academia and professionals have been suggested by authors to reduce the gap between the different areas of accounting impacts (e.g., [11]).

Despite the relative openness of the journals to new topics, which were identified, the findings also indicate a certain resilience of traditional themes, even when new methodologies and trending topics are applied. As a reference, the most cited papers from the first period seem to indicate that the underlying research themes remain. Furthermore, accounting research tends to incorporate new topics at relatively slow rates or in later stages than day-to-day practice, technical discussions, technological impacts, or regulation issues. This can be potentially explained by the necessary time to collect and assess data that underlies the different research methods in accounting. Notwithstanding, this might contribute to the feeling that accounting research has not been following the recent advances seen in the accounting profession.

In the first decade of this millennium, the initial focus on the AIS, which included the Internet and its potential resources (web-based services and reporting on the Internet), and related technologies, such as the ERP, was replaced. New challenges in accounting took place and then research was dedicated to new topics such as innovative ways of recording transactions and events, new business models, more efficient and effective methods for processing information, and, finally, new value chains, channels, and methods for disseminating useful information to a wider number of users. Then, the newest research appeared with a more diverse scope, covering new trends in accounting reporting overall, which includes new requirements, methods, and reporting channels, such as social media, as well as the different uses, risks, and benefits of emerging technologies for reporting and fraud detection. Case studies seem to be a relevant research method in the latest period assessed, mitigating a gap in the accounting area highlighted, for instance, by Kroon et al. [14].

Therefore, literature has added increasingly sophisticated data sources and analysis technologies, based on artificial intelligence, machine learning processes, big data, business intelligence, data mining, and data analytics, only to name the most recurrent ones. Despite the existence of literature reviews on the so-called emerging technologies, bibliometric studies on recent trends in AIS for the latest years have not yet been identified at this stage. Notwithstanding, the findings from this research are aligned with those recently released by Kumar [13], Kroon et al. [14], and Chiu et al. [15], which demonstrate a still broad potential for exploration of research in emerging technologies in accounting.

Manetti et al. [141], in this sense, highlight the potential use of emerging technologies to develop new paths for accounting. Furthermore, Kumar [13] identified a relative absence of research on specific areas, such as tax, governmental and nonprofit accounting. Notwithstanding, the findings from this paper, which identify an increasing relevance of studies on tax issues, particularly from the evidence of the latest period, should be highlighted. Moreover, the latest research has been incorporating and linking emerging technologies with trending topics in accounting, such as sustainability (nonfinancial information) reporting and other areas within social sciences. The latter includes research that uses psychology and sociology knowledge to amplify the scope of studies aiming to understand the different factors that may affect human behaviours, particularly when facing data assessment and decision-making processes.

Regardless of the shreds of evidence from this paper, the overall conclusions indicate that there is still an abundant scope for the development of research and use of emerging technologies in the accounting profession and education, which can be even broader from the perspective of accounting as a social science with an applied nature. In this context, its application in neurosciences, proposed by Tank and Farrell [12], is an example of a still-open challenge in accounting.

*3.1. Research Limitations*

The analysis performed in this paper, despite covering a relevant period, is limited to a single journal. Therefore, this limitation must be considered in the context of those conclusions, given that the decisions taken by its editorial board over this period may be seen as a relevant and subjective factor of neither measurable nor visible influence.

Additionally, the analysis was restricted to the abstracts found in those papers and, consequently, assumes the overall quality of the contents included in this piece of information. This limitation allows the assessment of a more relevant number of papers, despite only providing an overview of their contents. Exploring further sections would provide a more in-depth analysis of the papers' contents. Nonetheless, the decision making by researchers must consider the balance between the costs and the benefits of the analysis, considering its objectives and the level of information to be assessed.

Finally, as a qualitative approach, this research has a more particular and unavoidable constraint related to the judgments that should be made by research over the different steps of the analysis processes. The use of textual analysis tools may mitigate this limitation, but it necessarily remains, for instance during the choice of the most proper stop words for the analysis of the most frequent words.

*3.2. Future Avenues in Accounting and Information System Research*

Despite its limitation, this research may provide a relevant contribution to various interested parties in the accounting area, including academic members and professionals, as it identifies an overall perspective of the evolution trends and recent topics under discussion in this field.

The issues identified as trending topics can be seen as new sources of themes for researchers, and the results of these investigations can be reverted to benefits for academia and, consequently, professionals in the accounting area. The challenges professionals face needs to be monitored by researchers and educational institutions since it is crucial to include information technologies as a basic subject of curricula content for professional success. Additionally, professional associations can be of great usefulness whenever deficiencies or constraints are identified, as they possess the resources or can exert political pressure to overcome them.

Currently, the development of innovative research has also been combining the newest available data sources for data collection, technological methods for data assessment, and the so-called trending topics in accounting. Those topics include not only matters on international regulation and the sustainable perspective in accounting, such as the environmental, social and governance issues, but also new methods, channels and processes for improving the entities' auditing and reporting. Therefore, this seems to be a path to follow as an avenue for future research in accounting.

As an avenue to develop a literature review on AIS research and to go beyond the textual analysis proposed in this paper, we suggest the use of innovative tools, based on emerging technologies, that cover a more diversified set of journals dedicated to the use of new technologies in different accounting areas.

**Author Contributions:** This study was conceptualized, developed, written, and revised by F.A. and P.G.D.S. equally. All authors have read and agreed to the published version of the manuscript.

**Funding:** This research was supported by the Instituto Politécnico de Lisboa (IPL/2022/REPUK RAINE_ISCAL).

**Data Availability Statement:** The data source used for this research was collected from the Scopus database, freely available information to researchers. Nonetheless, for any questions or requests regarding the outputs from this paper, please contact the corresponding author.

**Conflicts of Interest:** The authors declare no conflict of interest.

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
