# Peer review of "Recent Trends in Accounting and Information System Research: A Literature Review Using Textual Analysis Tools"

_fintech, doi:10.3390/fintech2020015_

Round 1
Reviewer 1 Report
This paper is a decent literature analysis in accounting and information systems. Methodological approach is quite acceptable maintaining the interest of the readers. Nonetheless, applying the same analysis in more than one journals (despite that "the journal may have a particular editorial policy that can bias the proposed analysis") would significantly enhance paper's contribution to the field due to comparable findings and outputs. Yet, the paper justifies its publication with minor changes that are related with adding a subsection of research limitations (that should definitely justify in detail "the journal may have a particular editorial policy that can bias the proposed analysis") and future research proposals. Language is excellent. Well done!
Reviewer 2 Report
Thank you for allowing me to review this paper. However, I have shared my viewpoints.
In the abstract part, the findings are not clear.
The author(s) must highlight the fundamental objective in the abstract.
The introduction needs to improve, mainly focusing more on the justification for using bibliometric analysis.
The methodology seems OK.
The analysis part is OK.
The discussion section seriously needs to be improved. You need to state your main findings clearly, and it is a must to show where an opportunity to work "Accounting and Information Systems" is in future. Besides, it would be best if you focused on the "Discussion of the Findings". It would be best if you discussed your own findings compared to the results of other researchers. If you find something completely new, you need to explain it adequately.
Before concluding, you must focus on the "Contribution or Implication" of your research.
After the conclusion, "limitations and areas for future research" needs to include.
Regards.
Reviewer 3 Report
This paper applies freely available word analysis tools to words from journal papers published in a single journal, IJAIS, from 2000-2022. I’m unclear if the authors analyzed the full papers or just the abstracts as they write “restricted to papers with available abstracts.”
The methods the authors perform are simple, just counts of papers and citations by decade, and then the use of CountWordsFree and Lexos. In Figures 2-4 they provide word clouds, in Tables 1-3 they provide the most commonly used words. It seems they’ve just used these online tools and haven’t really performed any analysis of their own. They offer some qualitative commentary, such as the emergence of new terms, "XBRL", "mining" and "cloud", but they don’t analyze anything quantitatively. They don’t even present some basic analysis such as, numerically, which words changed their % prevalence in these papers the most over time.
I also don’t find the topic interesting, but perhaps I’m biased as I am not an accountant. I think to be acceptable for publication, the authors would have to do much more. Analyze a wide range of journals. Show how search terms are changing over time with some original quantitative analysis. Comment in more detail about which topics remain the same over time, which less so. Your word cloud figures contain quite obvious words like “accounting” - perhaps try and filter out many more “obvious” words that would feature in any accounting journal paper. Try and gain legitimate insight in how the accounting literature is changing.
I have an idea. Compare the journals with Google Trends. Maybe the journals are lagging behind the "hottest new topics" in accounting and you can show that quantitatively through original analysis.
The paper also has some minor errors. Figure 3 contains Information, but you said that was a stop word for both CountWordsFree and then you incorporated that into Lexos. Figure 1: write Number of papers not NoP, NoC. What is “Average”? Elsevier is Dutch, not based in the United States.
And finally, there are numerous typos and grammatical errors. A number of sentences in the paper are quite hard to read.
“The findings indicate that this journal is accessible to the newest discussions and topics.” What does this mean?
“The literature review has identified” - which literature review? Unclear.
“Accounting Information Systems” - decapitalize.
Your description of the coming subsections is quite confusing, “The next section is divided into two subsections….” instead just number the sections and state what each section does.
“the CountWordsFree and {L}exos” delete “the”.
“before mentioned”->aforementioned.
“It can be seen from those figures a trend…” awkward word order.
“which is also valid even for the last three-year period since it already indicates about or more than one-third of the figures for the same indicators published in the previous one.” Awkward language, “indicates” seems like the wrong word, and then you repeat indicators.
Round 2
Reviewer 2 Report
Dear Author(s)
Thank you for the improvement of your manuscript. Some of the sentences are very difficult to understand in the methodology part. Please try to make these simple.
Thanks.
Author Response
Dear reviewer,
Thank you so much again for the suggestion. The methodology was significantly reviewed to simplify and clarify the options taken.
Kind regards.
Reviewer 3 Report
Thanks to the authors for their revisions. It seems the other reviewers like the paper more than I do, so I'll defer to the majority.
Before the paper is finalized, could you please provide me with a clean (non-marked) manuscript? This manuscript is full of bold underlined red text and very hard to read.
Also, please change Figure 1, as I said in round 1. Please write Number of papers, Number of citations, not NoP and NoC. Also, what is "Average"? Does that mean average citations per paper?
Author Response
Dear reviewer,
Thank you again for your comments. Figure 1 was changed accordingly. I had improved the explanation regarding the average. Nonetheless, I changed it to "average citations". I am going to attach a clean version. Please allow me to suggest the following options to check the text with no marks: "On the Review tab, tap Display for Review. Tap the option you want: All Markup (inline) shows the final document with tracked changes visible inline. No markup shows the final document without tracked changed."
Round 3
Reviewer 3 Report
OK I think the paper is now fine.
Author Response
Thank you so much for all your suggestions!